# Prevalence of Antibiotic-Resistant *Escherichia coli* Isolated from Swine Faeces and Lagoons in Bulgaria

**DOI:** 10.3390/antibiotics10080940

**Published:** 2021-08-04

**Authors:** Lyudmila Dimitrova, Mila Kaleva, Maya M. Zaharieva, Christina Stoykova, Iva Tsvetkova, Maya Angelovska, Yana Ilieva, Vesselin Kussovski, Sevda Naydenska, Hristo Najdenski

**Affiliations:** 1The Stephan Angeloff Institute of Microbiology, Bulgarian Academy of Sciences, 26 Akad. G. Bonchev Str., 1113 Sofia, Bulgaria; milakalevavet@abv.bg (M.K.); zaharieva26@yahoo.com (M.M.Z.); christinastoikova98@abv.bg (C.S.); i.likovska@abv.bg (I.T.); mgatzovska@gmail.com (M.A.); illievayana@gmail.com (Y.I.); vkussovski@gmail.com (V.K.); hnajdenski@gmail.com (H.N.); 2University Multiprofile Hospital for Active Treatment Alexandrovska, Medical University, 1 Georgy Sofiiski Str., 1431 Sofia, Bulgaria; sevda.naydenska@abv.bg

**Keywords:** *Escherichia coli*, swine faeces and lagoons, antibiotic resistance, PCR methods, Phoenix M50, biofilms

## Abstract

Antimicrobial resistance (AMR) is a worldwide health problem affecting humans, animals, and the environment within the framework of the “One Health” concept. The aim of our study was to evaluate the prevalence of pathogenic strains of the species *Escherichia coli* (*E. coli*), their AMR profile, and biofilm-forming potential. The isolated strains from three swine faeces and free lagoons (ISO 16654:2001/Amd 1:2017) were confirmed using Phoenix M50 and 16S rDNA PCR. The antibiotic sensitivity to 34 clinically applied antibiotics was determined by Phoenix M50 and the disc diffusion method, according to the protocols of the CLSI and EUCAST. We confirmed the presence of 16 *E. coli* isolates, of which 87.5% were multi-drug-resistant and 31.25% performed strong biofilms. The possibility for the carrying and transmission of antibiotic-resistance genes to quinolones (*qnr*), aminoglycosides (*aac*(3)), β-lactamase-producing plasmid genes *amp*C, and *bla*SHV/*bla*TEM was investigated. We confirmed the carrying of *bla*SHV/*bla*TEM in one and *amp*C in seven isolates. The strains were negative for the virulence genes (ETEC (LT, STa, and F4), EPEC (*eae*), and STEC/VTEC (*stx* and *stx*2all)). The results should contribute to the development of effective measures for limitation and control on the use of antibiotics, which is a key point in the WHO action plan.

## 1. Introduction

Antimicrobial resistance (AMR) is a global public health challenge mainly caused by the widespread use of antibiotics in human and veterinary medicine for over 60 years. It is reported that deaths caused by AMR could increase from 700,000 in 2014 to 10 million by 2050 [1]. 

The use of antibiotics in animal husbandry has triggered the emergence and dissemination of antibiotic-resistant bacteria and genes for antibiotic resistance (GAR) from livestock farms to the surrounding environment. Different fractions of animal waste, such as swine manure or wastewater, are routinely applied to the fertilization of agricultural land in many countries. However, this waste has become a reservoir of resistant bacteria and various antibiotic residues that remain active in the soil from 20–30 to 40–60 cm depth for long periods of time via long-term manure application and transfer into groundwater by lagoons [2,3]. Their presence exerts a selective pressure on microorganisms and changes the microbial communities through the elimination of sensitive strains and increases the chances of survival for those containing GAR [4,5]. Resistant bacteria and their genetic determinants, such as plasmids, transposons, integrons, and genetic islands, can be spread and exchanged in different ways [6]. When bacteria come into contact with others, they exchange genetic determinants of resistance through horizontal gene transfer [7]. The path of distribution by transposons and plasmids between pathogenic and non-pathogenic bacteria in the environment, including GAR, is the most common along the food chain and the most important for the transmission of genetic variability [8,9]. Therefore, in addition to measures to reduce the use of antibiotics in animal husbandry, veterinary, and medical clinical practice, the prevalence of AMR and the important responsible genes as a risk factor for human and animal health should be investigated.

The species *Escherichia coli* (*Enterobacteriaceae*; *E. coli*) are part of the normal intestinal flora in humans and animals but often lead to urinary and gastrointestinal tract infection, hemolytic-uremic syndrome (HUS), sepsis, surgical site infection and meningitis. There is a lot of information in the literature, but in general, currently, over 171 somatic (O), 55 flagellar (H), and 80 capsular (K) antigens and 160 serological strains are known [10]. The most pathogenic *E. coli* strains are divided into several groups: enterohemorrhagic (EHEC), enterotoxigenic (ETEC), enteropathogenic (EPEC), Shiga toxin (verotoxin)-producing (STEC/VTEC), avian pathogenic (APEC), etc. Newborn and weaned animals are particularly susceptible to enteric colibacillosis, which involve ETEC and EPEC [11] due to their genetic immunodeficiency at birth [12]. The ETEC strains, which cause post-weaning diarrhea, produce heat-stable toxins a and b (STa and STb, respectively), which induce water and electrolyte loss from the intestine, and/or heat-labile enterotoxins (LT) [13,14]. The mechanism of action of STa and STb in newborns whith colibacillosis has been studied. STa stimulates the cyclic guanosine monophosphate (cGMP) production in the intestinal epithelial cells, which leads to electrolyte eand fluid secretion [15]. STb induces the duodenal and jejunal secretion of water and electrolytes by an uptake of Ca^2+^, leading to intoxication by Na^+^ and Cl^−^ accumulation and stimulates bicarbonate (HCO^3−^) secretion in cells. LT promotes the adherence, activates adenylyl cyclase in the basolateral plasma membrane of intestinal epithelial cells and leads to hypersecretion of electrolytes and water causing dehydration [16]. The fimbriae F4 occurs in both newborn and weaned animals with diarrhea [11]. The F4 ETEC strains colonize the length of jejunum and ileum [15]. The diarrhea caused by ETEC is usually watery with a characteristic yellowish, grey, or slightly pink color and smell [11]. The EPEC strains possess intimin Eae (outer membrane protein), which is responsible for the bacterial attachment to the host intestinal epithelium, which together with a complex secretion system (type III) leads to lesion formation [17]. The STEC/VPEC and certain EHEC strains produce different Shiga toxins (also called verotoxins), such as Stx1 (VT1), Stx2 (VT2), and caused hemorrhagic colitis in animals, and HUS in humans by contact [18]. There was an increasing trend in 2014–2018 of STEC human infections, which were the third most commonly reported zoonosis in the EU [19]. The EARS-NET reported the weighted mean percentages for third-generation cephalosporin resistance and aminoglycoside resistance in the population, as well as for combined resistance to three key antimicrobial groups (fluoroquinolones, third-generation cephalosporins, and aminoglycosides) [20]. 

Unfortunately, the data for Bulgaria are too scarce, as timely and constant monitoring is not applied in the areas around swine farms by responsible organizations. In addition, no systematic study has been conducted in Bulgaria on the prevalence of *E. coli*, their resistance to antibiotics, the prevalence of GAR in them, and biofilm formation capacity until now. The results will contribute to solving a global problem caused by the uncontrolled treatment of animals with antibiotics. Therefore, for the first time, our team investigated the path of distribution of the food-borne pathogenes *E. coli* and their GAR from faeces and lagoons. 

## 2. Results

### 2.1. Isolation of Single Bacterial Cultures

We isolated a total of 28 single colonies (15 with 3 lagoons and 13 with 3 faeces on CHROMagar CCA). This media detected and enumerated β-glucuronidase-positive *E. coli* (metallic blue to violet) and other coliforms (pink to red), according to ISO 9308-1. From them, 17 colonies were suspected for *E. coli*, and 11 colonies were suspected for coliforms. In the present study, we used only colonies, which were suspected for *E. coli* for subsequent experiments (Figure 1).

### 2.2. Biochemical and 16S rDNA Characterization

All 17 suspected for *E. coli* colonies were positive for indole and were confirmed after biochemical identification by BD Phoenix M50. In addition, 16S rDNA *E. coli* characterization by PCR (Figure 2) was carried out. We detected *E. coli* in both total 16S rDNAs from faeces and lagoons (Figure 2A) and from single colonies (Figure 2B,C).

Interestingly, isolate F2.1 was negative for *E. coli* by 16S rDNA detection. Therefore, our studies continued the study on other bacterial strains with proven genetic information.

### 2.3. Test for Biofilm Formation

We investigated the possibility of biofilm formation of all confirmed strains.

From all the tested 16 isolates (100%), 5 strains formed strong biofilms (31.25%), 6 strains formed moderate biofilms (37.5%), 4 strains formed weak biofilms (25%), and 1 strain did not form a biofilm (6.25%).

### 2.4. Antibiotic Resistance

According to the results obtained from BD Phoenix M50 (Table 2), 11 of the total 16 isolates were resistant to ampicillin (68.75%), trimethoprim (31.25%), and even the combination between trimethoprim and sulfamethoxazole (18.75%). Only one of them (L1.3; 6.25%) was also resistant to gentamicin, amoxicillin/clavulanic acid, ciprofloxacin, and colistin. Interestingly, this strain formed a strong biofilm (Table 1).

In addition, we performed an antibiotic disc diffusion test (Table 3). We found the resistances to amoxicillin (75%), tetracycline and chloramphenicol (56.25%), trimethoprim/sulfamethoxazole (43.75%), doxycycline hydrochloride (37.5%), and nalidixic acid (25%) in all the 16 isolates. The results for the resistance to ampicillin (68.75%) from BD Phoenix M50 were confirmed. Strain L1.3 was resistant also to pefloxacin. The resistances of two lagoon isolates (L1.4 and L3.4) to streptomycin were found. Isolate L1.4 formed a moderate biofilm, while isolate L3. did not (Table 1).

### 2.5. Detection of Antibiotic Resistance Genes

From a total of 12 tested strains resistant to β-lactam antibiotic, only L1.3 carried *bla*TEM and *bla*SHV β-lactam-resistance genes. The presence of *amp*C β-lactamases gene in three isolates from pigs for fattening and in four isolates from lagoons was found (Figure 3).

## 3. Discussion

The resistance in *E. coli* to some of the most widely used antibiotics in medical practice for the treatment of urinary tract infections, such as fluoroquinolones and sulfonamides, is a global problem. Due to the rapid spread of GAR, the treatment of urinary tract infections is ineffective in more than 50% of patients. According to a World Health Organization (WHO) report in 2017, most European Antimicrobial Resistance Surveillance Network (EARS-Net) countries have resistance between 10% and 25%, while more than 25% are found in Bulgaria, Cyprus, Italy, and Slovakia. Among Central Asian and European Surveillance of Antimicrobial Resistance (CAESAR) countries, the reported resistance exceeds 50% (Montenegro, Russia, Northern Macedonia, and Turkey), while in Serbia it varies between 25% and 50% [21]. The EARS-Net data show statistically significant increase in resistance of *E. coli* isolates in the EU to third generation cephalosporins [22]. The emergence of carbapenem-resistant *E. coli* has recently been identified, which is a serious challenge. Although the proportions of resistance are still low in Europe (around 1% and more), there is a tendency to increase worldwide [21,22]. A major mechanism of cephalosporin resistance is the production of β-lactamases, which hydrolyzes the β-lactam ring and inactivates β-lactam chemotherapeutics. The genetic determinants of resistance demonstrated in *E. coli* isolates include extended-spectrum β-lactamases (ESBL) encoded by various plasmid genes (*bla*SHV, *bla*CMY-2, *bla*TEM, etc.), as well as a number of GARs for quinolone resistance (*qnr*), trimethoprim (*dhf*), aminoglycosides (*aac*(3)), etc. [23].

Studies in Bulgaria showed that resident strains of *E.*
*coli* have a clear phenotypic and genotypic resistance profile to chemotherapeutics used in animal husbandry and human medicine. A high percent of resistance among pathogenic *E. coli* strains isolated from pigs in 2010–2015 was observed, which is a very alarming fact. In comparison with a previous study conducted in 2000–2004, the resistances to tetracycline antibiotics, streptomycin, spectinomycin, ampicillin, and sulfonamides, have increased by twofold. The percentage of resistance strains to ciprofloxacin has increased by tenfold during the same period. The widespread distribution of resistant *E. coli* strains has also been demonstrated in isolates, representative of the resident intestinal microflora of healthy pigs. The results showed the prevalence of *aad*A1 genes for determining streptomycin/spectinomycin resistance, *tet*(A) for determining tetracycline resistance, and *str*A/*str*B for determining streptomycin resistance. There is a limit information about the presence of the genes *sul*1 and *sul*2, *bla*TEM, and *tet*(B), which determine the bacterial resistances to sulfamethoxazole, aminopenicillins and cephalosporins of first generation, tetracyclines, respectively and the *int*l gene, which is responsible for the synthesis of the integrase enzyme from class 1 integrons. The lowest distribution is for the *aac*C2 gene, which is responsible for the resistance to gentamicin, kanamycin, tobramycin and netilmicin [24]. These facts support the hypothesis that the horizontal transfer of GAR in MDR commensal gut bacteria is one of the important risk factors for gene transfer, mainly through foodstuffs of animal origin or from the environment to humans.

Fluoroquinolone-resistant *E. coli* in China and Korea have been isolated from fecal samples [25,26]. From 171 samples isolated in 2015, from which 52 (30.4%) were from diseased swine and the other 119 (69.6%) were from healthy swine, a total of 59 *E. coli* isolates (34.5%) were confirmed as fluoroquinolone-resistant (21 (40.4% from diseased swine) and 38 (31.9% from healthy swine)). The researchers found plasmid-mediated quinolone-resistance (PMQR) genes in 9 isolates (15.3%) and efflux pump activity in 56 isolates (94.9%). The authors reported that the resistance to fluoroquinolones has increased significantly in swine compared to in previous studies in Korea, although fluoroquinolones have been banned as a feed additive since 2009. These authors investigated the *qnr*A and *qnr*B genes, but as in our study, they did not prove them [25]. All isolates from China showed the moderate rates of the resistance to norfloxacin (43.0%), ciprofloxacin (47.6%), ofloxacin (47.0%), and levofloxacin (38.8%). They also did not detect the presence of *qnr*A and *qnr*B genes [26]. Hu et al. (2017) suggested that the predominant PMQR genes detected in human isolates were *qnr*A and *qnr*B, whereas *qnr*S was detected in swine samples. Probably for this reason, we also failed to confirm any of these two genes (*qnr*A and *qnr*B) (Figure 3).

According to a study by the National Diagnostic Research Veterinary Medical Institute (NDRVMI) and University of Forestry in Bulgaria conducted in the period 2012–2014, the number of positive strains of *E. coli* from all samples isolated from different pig farms in the country ranged between 50% and 70%. Above 75% of them were resistant to amoxicillin and erythromycin and from 51% to 75% of them were resistant to ampicillin, oxytetracycline, thiamulin, streptomycin, doxycycline, tylosin, and tilmicosin. It has been found that sensitivity is most strongly established to non-use agents (amikacin, cefquin, and cefotaxime), less commonly used in practice (kanamycin), or new agents in the fluorinated quinolone groups (ciprofloxacin, enrofloxacin, and pefloxacin) and amphenicols [27].

We performed antimicrobial susceptibility tests against additional eight classes of drugs and six other antibiotics (a total of 34 antimicrobial agents). From all isolated *E. coli* strains, 87.5% of them are MDR (only F1.4 and L3.1 were sensitive to the antibiotics used). Only 18.75% of all isolated *E. coli* strains were resistant to aminoglycosides (L1.4 and L3.4 were resistant to streptomycin, and L1.3 was resistant to gentamicin), 81.25% of them were resistant to penicillins, 25% of them were resistant to fluoroquinolones, 6.25% of them (only L3.2) were resistant to macrolides, and 68.75% of them were resistant to other antibiotics. The isolated *E. coli* strains from swine faeces and lagoons were susceptible to monobactams, cephalosporins, and carbapenems. We found high resistance to *β*-lactam (amoxicillin and ampicillin) and tetracycline (tetracycline and doxycycline hydrochloride) antibiotics (Table 2 and Table 3). The percentage of resistance was also high. Moreover, compared with a previous study in Bulgaria, the current research demonstrates a substantial increase in resistance to trimethoprim/sulfamethoxazole from 7.1% in the period 2012–2014 [27] to 43.75% in our study and a significant increase in resistance to nalidix acid from 11.1% in the period 2012–2014 [27] to 25% that we found. The ampicillin and amoxicillin resistance are found today (70–80%), including those to gentamicin and pefloxacin (about 4–7%) [27]. Decreased resistances to doxycycline (from 64.7% in the period 2012–2014 [27] to 37.5% in our case), streptomycin (from 63.1% in the period 2012–2014 [27] to 12.5% found by us), erythromycin (from 80% in the period 2012–2014 [27] to 6.25%) have been observed. Dimitrova et al. (2016) also documented that the isolated *E. coli* strains were susceptible to amikacin, cefotaxim, ciprofloxacin and norfloxacin [27]. Urumova (2016) studied the AMR in *E. coli* in the period 2012–2016 from different regions in Bulgaria (Shumen, Ruse, Razgrad, Yambol, and Varna). She found high resistance in growing pigs, compared with that in suckling pigs, fattening swine, and lagoons [24]. Compared with her study, it was found that the resistances to ampicillin (68.75%) and amoxicillin/clavulanic acid (from 2.2% in the period 2012–2016 to 6.25% in our study) were doubled and tripled, respectively [24]. The researcher also proved high resistance to streptomycin (69.4%) [24], as reported by previous authors [27]. A slight resistance to ciprofloxacin (5.2%) was observed, which was also confirmed by our results (6.25%). The resistance to tetracycline was almost not changed from 73.3% in the period 2012–2016 [24] to 56.25% found by us, indicating that the antibiotic continues to be used in swine farms. Only one isolate out of a total of 157 growing animals was found was to be resistant to ceftazidime and cefotaxime [24].

## 4. Materials and Methods

### 4.1. Swine Farm and Sample Collection

The study included a swine farm near Kostinbrod, which was founded in 2008 and today is a part of the Hog and Pig Farming Companies in Bulgaria. It was designed for up to 8200 breeding animals and their offspring. It was built on an area of about 70.5 decares with introduced biosecurity measures. All normative documents for protection and animal welfare were observed. Three samples from pig faeces (F1–F3) and three samples from lagoons (L1–L3) were collected in March 2020 according to ISO 5667-3:2018. Probes F1 and F2 were from pigs for fattening, and F3 was from mother pigs.

### 4.2. Isolation of Single Bacterial Cultures

Single colonies, suspected for *E. coli*, were isolated by ISO 16654:2001/Amd 1:2017 and ISO 9308-1 with some modifications. Enriched samples from faeces and lagoons were cultured on HiCrome™ Chromogenic Coliform Agar (CCA) (M1991I, HiMedia, Mumbai, India) at 41 °C for 24 h. For positive controls, we used *E. coli* ATCC 35218 (American Type Cell Culture Collection, Manassas, VA, USA) and *E. coli* O:157 (Collection of the Stephan Angeloff Institute of Microbiology). All isolated colonies were morphologically characterized by automatic HD colony counter Scan 1200 (INTERSCIENCE, Saint-Nom-la-Bretèche, France).

### 4.3. Biochemical Characterization

All isolated single colonies suspected for *E. coli* were tested for indole production from trypthophan deamination using Kovacs’ Indole Reagent (R008, HiMedia, Mumbai, India). We used automatic BD PhoenixTM M50 system (443624, Becton, Dickinson and Company, Franklin Lakes, NJ, USA) for a full biochemical characterization of isolates by the laboratory procedure, as described by the manufacturer. Briefly, the bacterial colonies (0.5 McFarland) were inoculated into the ID broth (246001, Becton, Dickinson and Company, Franklin Lakes, NJ, USA), and 25 µl were transferred into the AST broth (246003, Becton, Dickinson and Company, Franklin Lakes, NJ, USA) with one drop of an AST indicator solution (246004, Becton, Dickinson and Company, Franklin Lakes, NJ, USA). The suspensions were poured in NMIC/ID-76 panels for Gram-negative bacteria (448103, Becton, Dickinson and Company, Franklin Lakes, NJ, USA) and loaded into the instrument at 35 °C for 24 ± 4 h. The obtained data were analyzed by EpiCentre™ software (V7.45A/V6.71A). The minimum inhibitory concentration (MIC) was determined according to CLSI guidelines.

### 4.4. Isolation of 16S rDNA

The total rDNA was extracted from faecal and lagoon samples with the GeneMATRIX Stool DNA Purification Kit (E3575, EURx Ltd, Gdańsk, Poland). The rDNA concentration and purity were determined with NanoDrop 1000 (Thermo Fisher Scientific Inc., Waltham, MA, USA) by migration in 0.8% SeaKem^®^ LE Agarose gels (50004L, Lonza Group Ltd, Basel, Switzerland) in a 1× TBE buffer. The 16S rDNA from the confirmed *E. coli* was isolated with the GenEluteTM Bacterial Genomic DNA Kit (NA2120, Merck (Sigma-Aldrich, St. Louis, MO, USA).

### 4.5. PCR Analysis

The extracted total 16S rDNAs from faeces and lagoon samples were subjected to conventional PCR with gene-specific primers for *E. coli*. The isolated 16S rDNAs from single colonies were subjected to conventional and multiplex PCR with primers linked to virulence and antibiotic resistance genes in the isolated *E. coli* strains (Table 4). For PCR amplification, we used the Taq PCR Master Mix (2×) protocol (E2520, EURx Ltd, Gdańsk, Poland) being optimized in our laboratory as follows: 1 cycle of initial denaturation running at 95 °C for 5 min; total 25 cycles of denaturation (at 94 °C for 30 s), annealing (depending on the temperature of the primer for 60 s) and extension (at 72 °C for 1 min); 1 cycle of final extension (at 72 °C for 7 min) and cooling (at 4 °C). The PCR products were visualized in 1.5% agarose gels. For positive controls, we used the following strains: *E. coli* O:157 for the detection of *E. coli* strains, *E. coli* ATCC 43887 for the detection of eae genes, *E. coli* ATCC 35401 containing LT, Enterococcus faecalis V347 DSM 8629 containing ermB, Enterobacter hormaechei subsp. xiangfangensis DSM 46348 containing ampC, Klebsiella pneumonia DSM 16609 containing blaSHV-5, *E. coli* ATCC 35218 containing *bla*TEM, and Citrobacter sp. DSM 30042 containing the *PMQR* gene (qnrB60).

### 4.6. Disc Diffusion Method

Antimicrobial susceptibility testing was performed according to the protocols of the CLSI [44]. The results were evaluated according to the EUCAST cut-off values [45] using antibiotics applicable to the treatment of patients, namely meropenem (10 µg, MEM10C) from Mast Group Ltd., UK, penicillin-G (10 U, SD028-1PK), ampicillin (10 µg, SD002-1PK), amoxycillin (25 µg, SD129-1PK), amoxycillin/clavulanic acid (20/10 µg, AUG30C), carbenicillin (100 µg, SD004-1PK), cefamandole (30 µg, SD200-1PK), erythromycin (15 µg, SD013-1PK), clarithromycin (15 µg, SD192-1PK), streptomycin (10 µg, SD031-1PK), tetracycline (30 µg, SD037-1PK), doxycycline hydrochloride (30 µg, SD012-1PK), chloramphenicol (30 µg, SD006-1PK), nalidixic acid (30 µg, SD021-1PK), ciprofloxacin (5 µg, SD060-1PK), pefloxacin (5 µg, SD070-1PK), and co-trimoxazole (25 µg, SD010-1PK) from HiMedia, India.

### 4.7. Test for Biofilm Formation

The ability of the isolated single colonies from pig faeces and lagoons to form biofilms was tested in flat-bottomed 96-well plates, according to the protocol of Stepanović et al. with small modification [46]. Briefly, we used a Brain Heart Infusion broth (M210, HiMedia, India) supplemented with 2% D-(+)-glucose (G7021, Merck, Darmstadt, Germany). The bacterial inoculums were cultured at 37 °C for 18 h, and then, the supernatants were aspirated gently. The cells were washed two times with 200 µl PBS and fixed with methanol (32213-M; Sigma-Aldrich, USA) for 15 min. The plates were placed to dry; each well was stained with a 200 µl 2% gentian violet solution for 5 min and was washed under running water. As a negative control, we used blank. The test was performed sixfold. The biofilms were photodocumented on microscopic-configuration Nikon Eclipse-Ci-L (Nikon Instruments Europe BV, Netherlands) and then dissolved with a 33% glacial acetic acid solution. The optical density (OD) was measured at 570 nm by using an ELISA reader ELx800 (BioTek Instruments, Winooski, Vermont, USA). We used the following classification of Christensen et al. (Table 5) to determine the adherence potential [47]:

## 5. Conclusions

From the total of 28 single colonies, 16 isolates (100%) were confirmed with BD Phoenix M50 and 16S rDNA PCR to be *E. coli*. The antimicrobial tests showed that most of them (87.5%) had MDR. Moreover, 31.25% of the *E. coli* strains were capable of forming strong biofilms. We found high percents of resistance varying between 50% and 75% to amoxicillin, ampicillin, tetracycline, and chloramphenicol. The resistances (25%–50%) to clinically administered antibiotics, such as trimethoprim, trimethoprim/sulfamethoxazole, doxycycline hydrochloride, and nalidixic acid, were no less. We proved the β-lactamase genes *bla*TEM/*bla*SHV in one isolate from lagoon and *amp*C in three isolates from pigs for fattening and in four isolates from lagoons.

From the results presented here and compared with the data for the period 2012–2016, high resistance to tetracycline was found in growing pigs and fattening swine, which is a worrying fact as coliforms resistant to this antibiotic may be ingested during consumption. This also applies to the antibiotics ampicillin and amoxicillin, which continue to be used in veterinary practice. Probably, less commonly applied are streptomycin, erythromycin, and doxycycline. Considering the clinical importance of antibiotic resistance emergence in veterinary and human medicine, the prescription of antibiotics should be carefully monitored and regulated, in order to reduce AMR in food industry in Bulgaria.

## Figures and Tables

**Figure 1 antibiotics-10-00940-f001:**
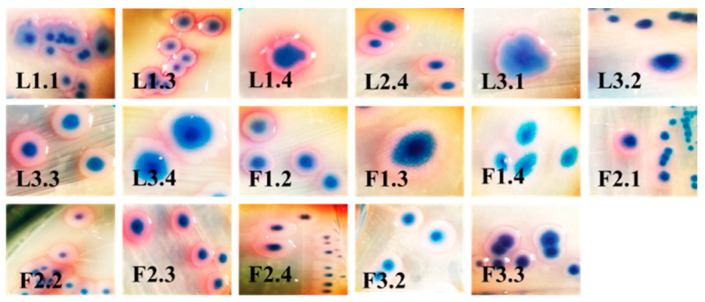
Morphological characteristics of bacterial strains, which were suspected for *Escherichia coli* (*E. coli*). Legend: L, lagoon; F, faeces; the first number, the number of probe; the second number, the number of isolate.

**Figure 2 antibiotics-10-00940-f002:**
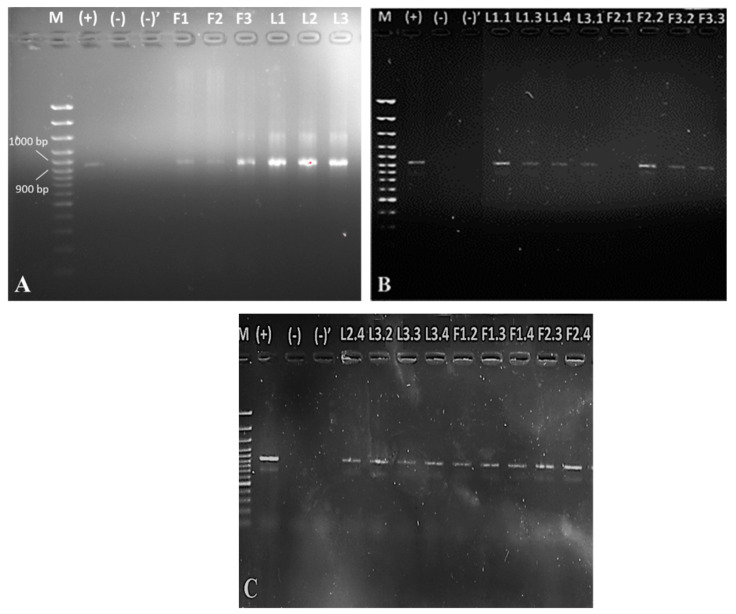
Detection of *E. coli* isolated from lagoons L1–L3 and faeces F1–F3 (**A**) and from single colonies (**B**,**C**) by 16S rDNA PCR. Legend: M, marker; (+), positive control (*E. coli* O:157 for 16S rRNA PCR identification); (−), control of purity in the place of dispensing MasterMix; (−)’, control of purity in the place of dispensing DNA.

**Figure 3 antibiotics-10-00940-f003:**
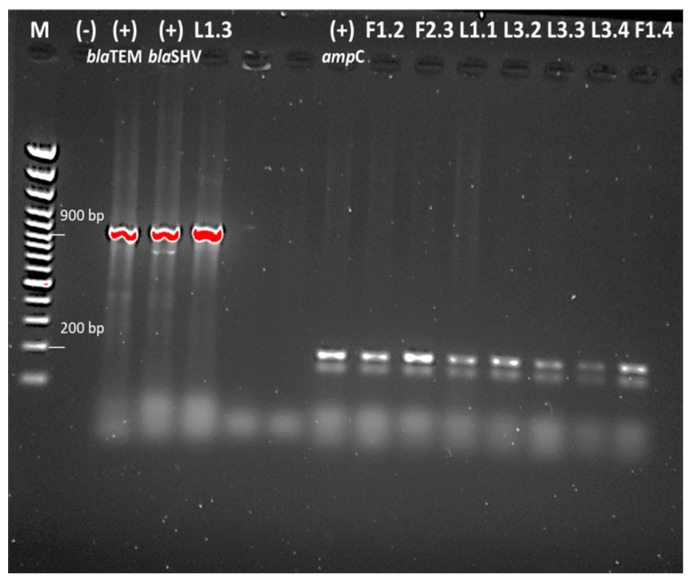
Detection of *amp*C β-lactamases gene and *bla*TEM and *bla*SHV β-lactam-resistance genes.

**Table 1 antibiotics-10-00940-t001:** Adherence of the isolated *E. coli* from lagoons and faeces, compared with those of the controls.

Strain	OD_570_ _nm_	Adherence	Biofilm	Strain	OD_570 nm_	Adherence	Biofilm
ATCC 35218	0.463	MA	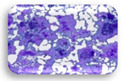	L3.4	0.158	NA	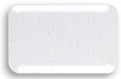
O:157	1.041	SA	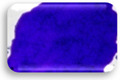	F1.2	1.083	SA	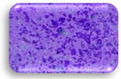
Blank	0.162	---	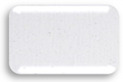	F1.3	0.261	WA	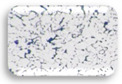
L1.1	0.361	MA	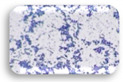	F1.4	0.261	WA	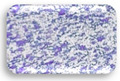
L1.3	1.103	SA	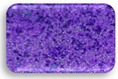	F2.2	0.358	MA	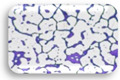
L1.4	0.364	MA	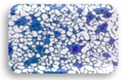	F2.3	1.183	SA	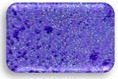
L2.4	1.222	SA	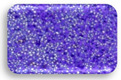	F2.4	0.378	MA	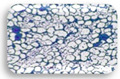
L3.1	0.361	MA	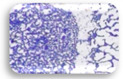	F3.2	0.204	WA	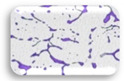
L3.2	0.382	MA	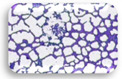	F3.3	0.328	WA	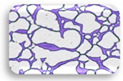
L3.3	1.188	SA	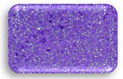				

Legend: NA, non-adherent confirmed *E. coli* strains; WA, weakly adherent confirmed *E. coli* strains; MA, moderately adherent confirmed *E. coli* strains; SA, strongly adherent confirmed *E. coli* strains.

**Table 2 antibiotics-10-00940-t002:** Antibiotic resistances of the isolates identified as *E. coli* by BD Phoenix M50.

Drug Class	Antibiotic/Strain	F1.2	F1.3	F1.4	F2.1	F2.2	F2.3	F2.4	F3.2	F3.3	L1.1	L1.3	L1.4	L2.4	L3.1	L3.2	L3.3	L3.4	*E. coli* ATCC 35218
Aminoglycosides	Tobramycin	S	S	S	S	S	S	S	S	S	S	S	S	S	S	S	S	S	S
	Amikacin	S	S	S	S	S	S	S	S	S	S	S	S	S	S	S	S	S	S
	Gentamicin	S	S	S	S	S	S	S	S	S	S	**R**	S	S	S	S	S	S	S
Penicillins	Amoxicillin/clavulanic acid	S	S	S	S	S	S	S	S	S	S	**R**	S	S	S	S	S	S	S
	Ampicillin	S	**R**	**R**	S	S	**R**	S	**R**	**R**	**R**	**R**	S	**R**	S	**R**	**R**	**R**	**R**
	Piperacillin/tazobactam	S	S	S	S	S	S	S	S	S	S	S	S	S	S	S	S	S	S
Monobactams	Aztreonam	S	S	S	S	S	S	S	S	S	S	S	S	S	S	S	S	S	S
Cephalosporins	Cefazolin	**I**	**I**	**I**	**I**	X	**I**	**I**	**I**	**I**	**I**	**I**	**I**	**I**	**I**	**I**	**I**	**I**	**I**
	Cefotaxime	S	S	S	S	S	S	S	S	S	S	S	S	S	S	S	S	S	S
	Ceftazidime	S	S	S	S	S	S	S	S	S	S	S	S	S	S	S	S	S	S
	Cefuroxime	**I**	**I**	**I**	**I**	**I**	**I**	**I**	**I**	**I**	**I**	**I**	**I**	**I**	**I**	**I**	**I**	**I**	**I**
	Cephalexin	S	S	S	S	S	S	S	S	S	S	S	S	S	S	S	S	S	S
Fluoroquinolones	Ciprofloxacin	S	S	S	S	**I**	S	S	S	S	S	**R**	S	S	S	S	S	S	S
Carbapenems	Ertapenem	S	S	S	S	S	S	S	S	S	S	S	S	S	S	S	S	S	S
	Imipenem	S	S	S	S	S	S	S	S	S	S	S	S	S	S	S	S	S	S
	Meropenem	S	S	S	S	S	S	S	S	S	S	S	S	S	S	S	S	S	S
Other agents	Colistin	X	X	X	X	X	X	X	X	X	X	**R**	X	X	X	X	X	X	X
	Fosfomycin	S	S	S	S	S	S	S	S	S	S	S	S	S	S	S	S	S	S
	Nitrofurantoin	S	S	S	S	S	S	S	S	S	S	S	S	S	S	S	S	S	S
	Trimethoprim	**R**	S	S	S	**R**	S	S	**R**	**R**	S	**R**	S	S	S	S	S	S	S
	Trimethoprim/sulfamethoxazole	**R**	S	S	S	**R**	S	S	S	S	S	**R**	S	S	S	S	S	S	S

**Table 3 antibiotics-10-00940-t003:** Antibiotic resistances by the disc diffusion method of the isolated *E. coli* strains.

Drug Class	Antibiotic/Strain	F1.2	F1.3	F1.4	F2.2	F2.3	F2.4	F3.2	F3.3	L1.1	L1.3	L1.4	L2.4	L3.1	L3.2	L3.3	L3.4	*E. coli* O:157	*E. coli* ATCC 35218
Tetracycline	Tetracycline	**R**	S	S	**R**	**I**	S	**R**	S	**R**	**R**	**R**	**R**	**I**	**R**	**R**	**I**	**R**	S
	Doxycycline hydrochloride	**R**	**I**	S	**R**	**R**	S	**R**	**R**	S	**R**	S	**R**	S	S	S	S	S	S
Macrolides	Erythromycin	S	**I**	S	**I**	S	S	**I**	**I**	**I**	**I**	**I**	S	**I**	**R**	S	S	S	S
	Clarithromycin	S	**I**	S	S	S	S	**I**	S	S	S	S	S	S	S	S	S	S	S
Cephalosporins	Cefamandole	S	S	S	S	S	S	S	S	S	S	S	S	S	S	S	S	S	S
Fluoroquinolones	Nalidixic acid	S	**R**	S	S	S	S	S	S	**R**	**R**	S	S	S	S	S	**R**	S	**R**
	Pefloxacin	S	S	S	S	S	S	S	S	S	**R**	S	S	S	S	S	S	S	S
	Ciprofloxacin	S	S	S	S	S	S	S	S	S	S	S	S	S	S	S	S	S	S
Penicillins	Ampicillin	**R**	**R**	S	S	**R**	S	**R**	**R**	**R**	**R**	S	**R**	S	**R**	**R**	**R**	S	**R**
	Amoxicillin	S	**R**	**R**	S	**R**	S	**R**	**R**	**R**	**R**	**R**	**R**	S	**R**	**R**	**R**	n.m.	**R**
	Amoxicillin/clavulanic acid	S	S	S	S	S	S	S	S	S	S	S	S	S	S	S	S	S	S
	Penicillin	S	S	S	S	S	S	S	S	S	S	S	S	S	S	S	S	S	S
	Carbenicillin	S	S	S	S	S	S	S	S	**I**	**I**	S	S	**I**	S	S	S	S	S
Carbapenems	Meropenem	S	S	S	S	S	S	S	S	S	S	S	S	S	S	S	S	S	S
Aminoglycosides	Streptomycin	S	S	S	S	S	S	S	S	S	**I**	**R**	S	S	S	S	**R**	S	**R**
Other agents	Chloramphenicol	**R**	**R**	S	**R**	**R**	S	**R**	**R**	S	S	S	**R**	S	S	**R**	**R**	**R**	**R**
	Trimethoprim/sulfamethoxazole	**R**	**R**	S	**R**	S	S	**R**	**R**	S	**R**	**R**	S	S	S	S	S	S	S

**Table 4 antibiotics-10-00940-t004:** List of primers used in this study with their sequences.

Primers	Sequences	Reference
*E. coli* 16S rDNA F	5′-AGA GTT TGA TCC TGG CTC AG-3′	[28]
*E. coli* 16S rDNA R	5′-CTT GTG CGG GCC CCC GTC AAT TC-3′	
*stx*1-1 F	5′-TTA GAC TTC TCG ACT GCA AAG-3′	[29,30]
*stx*1-1 R	5′-TGT TGT ACG AAA TCC CCT CTG-3′	
*stx*2all-1 F	5′-TTA TAT CTG CGC CGG GTC TG-3′	[30]
*stx*2all-2 R	5′-AGA CGA AGA TGG TCA AAA CG-3′	
LT1 F	5′-TTA CGG CGT TAC TAT CCT CTC TA-3′	[31]
LT2 R	5′-GGT CTC GGT CAG ATA TGT GAT TC-3′	
STa1 F	5′-TCC CCT CTT TTA GTC AGT CAA CTG-3′	[31]
STa2 R	5′-GCA CAG GCA GGA TTA CAA CAA AGT-3′	
F4-1 F	5′-ATC GGT GGT AGT ATC ACT GC-3′	[31]
F4-2 R	5′-AAC CTG CGA CGT CAA CAA GA-3′	
*eae* (Intimin)-1 F	5′-CAT TAT GGA ACG GCA GAG GT-3′	[30,32]
*eae* (Intimin)-2 R	5′-ATC TTC TGC GTA CTG CGT TCA-3′	
*qnr*A F	5′-GGG TAT GGA TAT TAT TGA TAA AG-3′	[33]
*qnr*A R	5′-CTA ATC CGG CAG CAC TAT TTA-3′	
*qnr*B F	5′-GAT CGT GAA AGC CAG AAA GG-3′	[34]
*qnr*B R	5′-ACG ATG CCT GGT AGT TGT CC-3′	
*aac*(3)-IV F	5′-CTT CAG GAT GGC AAG TTG GT-3′	[35]
*aac*(3)-IV R	5′-TCA TCT CGT TCT CCG CTC AT-3′	
*bla*SHV F	5′-TCG CCT GTG TAT TAT CTC CC-3′	[36]
*bla*SHV R	5′-CGC AGA TAA ATC ACC ACA ATG-3′	
*bla*TEM F	5′-TCG GGG AAA TGT GCG CG-3′	[37,38]
*bla*TEM R	5′-TGC TTA ATC AGT GAG GCA CC-3′	
*amp*C F	5′-AAT GGG TTT TCT ACG GTC TG-3′	[39,40]
*amp*C R	5′-GGG CAG CAA ATG TGG AGC AA-3′	
*amp*C F	5′-GTG ACC AGA TAC TGG CCA CA-3′	[41]
*amp*C R	5′-TTA CTG TAG CGC CTC GAG GA-3′	
*erm*B F	5′-GAA AAA GTA CTC AAC CAA ATA-3′	[42,43]
*erm*B R	5′-AAT TTA AGT ACC GTT AC-3′	
*erm*B F	5′-GCA TTT AAC GAC GAA ACT GGC T-3′	[41]
*erm*B R	5′-GAC AAT ACT TGC TCA TAA GTA ATG GT-3′	

**Table 5 antibiotics-10-00940-t005:** Correlation between the optical density of samples and bacterial adherence [47].

Formula	Adherence
OD_probe_ ≤ OD_blank_	non-adherent
OD_blank_ < OD_probe_ ≤ 2 × OD_blank_	weakly adherent
2 × OD_blank_ < OD_probe_ ≤ 4 × OD_blank_	moderately adherent
4 × OD_blank_ < OD_probe_	strongly adherent

## Data Availability

The data presented in this study are available on request from the corresponding author.

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
