# Peer review of "Prevalence of Antibiotic-Resistant Escherichia coli Isolated from Swine Faeces and Lagoons in Bulgaria"

_antibiotics, 2021, doi:10.3390/antibiotics10080940_

Round 1

Reviewer 1 Report

The authors investigated the antimicrobial resistance profiles of 16 strains of E. coli isolated from swine feces and lagoons in Bulgaria. With infections with MDR organisms increasing worldwide, areas where antimicrobial stewardship is not practiced well is important to identify. The authors highlight issues of antibiotics being too liberally used in veterinary and human medicine worldwide. The present study is interesting and it is important to have information widely available on antimicrobial resistance trends. The manuscript unfortunately is rife with grammatical and English language errors. Prior to publication it will need to be carefully reread and revised by the authors as many of the errors make it difficult to understand their science and claims. Furthermore, the information presented is somewhat niche with a relatively low number of strains isolated from a single farm that was only founded a little over a decade ago. Below are some of my other additional and specific concerns.

Abstract –

Sentence 2 – should be “the aim of our study. . . “

“We proved the presence” should rephrase and refrain from using “proved” in a scientific paper. Furthermore, the sentence is difficult to understand. Should specify where the presence of the E. coli isolates were “proved” to be. Please check manuscript for more instances of “proved”

Body -

Line 56 – what is @rst?..

Line 59 – another weird symbol. . I also do not understand this list at all.

Line 61 – just “normal” not normally. Human and animals should be plural

Line 73 – Whith? Newborn should be plural

Line 75 – lead pluralized, end?... should be and..

I am going to stop commenting on grammatical errors at this point of the manuscript. Please reread the entire manuscript very carefully for grammatical errors.

Lines 99-100 – is antimicrobial resistance and inappropriate antibiotic use a social problem?

Figure 3 – why were all isolates not tested?

Line 182 – it’s extended-spectrum not broad.

Line 191 – Increased twice? Do you mean two-fold or double?

Author Response

RESPONSES TO REVIEWER 1 COMMENTS

The authors investigated the antimicrobial resistance profiles of 16 strains of E. coli isolated from swine feces and lagoons in Bulgaria. With infections with MDR organisms increasing worldwide, areas where antimicrobial stewardship is not practiced well is important to identify. The authors highlight issues of antibiotics being too liberally used in veterinary and human medicine worldwide. The present study is interesting and it is important to have information widely available on antimicrobial resistance trends. The manuscript unfortunately is rife with grammatical and English language errors. Prior to publication it will need to be carefully reread and revised by the authors as many of the errors make it difficult to understand their science and claims. Furthermore, the information presented is somewhat niche with a relatively low number of strains isolated from a single farm that was only founded a little over a decade ago. Below are some of my other additional and specific concerns.

Response: Thank you so much for your important comments and specific concerns. The corresponding changes were marked in red in the text. The detailed responses to your comments could be found below.

Abstract –

Sentence 2 – should be “the aim of our study. . . “

 “We proved the presence” should rephrase and refrain from using “proved” in a scientific paper. Furthermore, the sentence is difficult to understand. Should specify where the presence of the E. coli isolates were “proved” to be. Please check manuscript for more instances of “proved”

Response: We acknowledge your comments. They have been corrected.

Body -

Line 56 – what is @rst?..

Line 59 – another weird symbol. . I also do not understand this list at all.

Response: Our apologies. А formatting error occurred and the text has been removed.

Line 61 – just “normal” not normally. Human and animals should be plural

Line 73 – Whith? Newborn should be plural

Line 75 – lead pluralized, end?... should be and..

I am going to stop commenting on grammatical errors at this point of the manuscript. Please reread the entire manuscript very carefully for grammatical errors.

Response: Our apologies. We reread and revised for grammatical errors the manuscript. This was corrected in the manuscript,

Lines 99-100 – is antimicrobial resistance and inappropriate antibiotic use a social problem?

Response: We acknowledge your comment. The problem is a global.

Figure 3 – why were all isolates not tested?

Response: We defined the other colonies as coliforms. Therefore, they were not used for further experiments. We added the explanation in the text (lines 104-106).

Line 182 – it’s extended-spectrum not broad.

Line 191 – Increased twice? Do you mean two-fold or double?

Response: We acknowledge your comments. We corrected them.

Reviewer 2 Report

In the Introduction in line 56-60 there are some formal mistakes in the text. Also this part has different line spacing and also in different parts of text.

In material and methods are animal samples described, but not how many samples of faces were collected.

Line 277 E. coli is without italic.

Only 28 isolates were identified? This number is not too much for one study.

Results and discussion are very well described but I miss some novelty of this study.

Can authors improve the novelty of this study?

Author Response

RESPONSES TO REVIEWER 2 COMMENTS

In the Introduction in line 56-60 there are some formal mistakes in the text. Also this part has different line spacing and also in different parts of text.

Response: Our apologies. А formatting error occurred and the text has been removed.

In material and methods are animal samples described, but not how many samples of faces were collected.

Response: This is noted in materials and methods. We collected 3 samples of faeces and 3 samples of lagoons from the swine farm. We made an additional clarification in the text.

Line 277 E. coli is without italic.

Response: Our apologies. It has been corrected

Only 28 isolates were identified? This number is not too much for one study.

Response: We accept your criticism. This investigation is part of a large-scale study that we plan to conduct in the territory of our country. Due to the COVID-19 pandemic in March 2020 and the closure of our country, samples were collected from only one pig farm, which allowed us. We assume that at the same time the farmers gave preventive drugs to the animals. That’s why the bacterial growth was weak and we succeeded to isolate only 28 colonies, from which 17 were suspected for E. coli and 11 – for coliforms.

Results and discussion are very well described but I miss some novelty of this study.

Can authors improve the novelty of this study?

Response: We are aware that the problem we have considered is quite popular for large countries such as Japan, Belgium, Sweden, China, etc., but for Bulgaria, which is a small country, our study is completely new and appropriate. That’s why we hope that you will give a positive assessment and our results will published, so they can become visible not only to our society, but also to all readers of the Journal Antibiotics. In addition to disseminating the results, we plan to make a site where all publications will be uploaded, including this one, thus contributing to increasing the readers of the Journal.

Reviewer 3 Report

The study deals with an up-to-date subject of concern, which is Antimicrobial Resistence (AMR). The authors made a deeply research about several antimicrobials disposed in lagoons and faeces from swine. It is a novel study, of interest because it matters to people and to animals. Most antibiotic are shared among people and animals provoking resistance problems. 

I attached a file with considerations about the manuscript. They are indicated and highlighted to improve the presentation itself. 

Congratulations on the article. I found it really interesting.

Author Response

RESPONSES TO REVIEWER 3 COMMENTS

The study deals with an up-to-date subject of concern, which is Antimicrobial Resistence (AMR). The authors made a deeply research about several antimicrobials disposed in lagoons and faeces from swine. It is a novel study, of interest because it matters to people and to animals. Most antibiotic are shared among people and animals provoking resistance problems. 

I attached a file with considerations about the manuscript. They are indicated and highlighted to improve the presentation itself. 

Congratulations on the article. I found it really interesting.

Response: We acknowledge your kind comment. All your recommendations are marked in red in the text.

Round 2

Reviewer 1 Report

I would like to thank the authors for the significant revisions of the manuscript. It is much improved from the last version. I believe the paper is fit for publication at this point save some minor revisions to English language.

Reviewer 2 Report

All comments were accepted.